# Determining Chemotherapy Agents in Saliva through Spectrometry and Chromatography Methods Correlated with Periodontal Status in Oncology Patients

**Diana Cristala Kappenberg Niţescu [1], Liliana Păsărin [1], Silvia Mârţu [1], Cornelia Teodorescu [1], Bogdan Vasiliu [1], Ioana Mârţu [2,†], Ionut Luchian [1,\*] and Sorina Mihaela Solomon [1]**

[1] Periodontology Department, Faculty of Dentistry, "Grigore T. Popa" University of Medicine and Pharmacy of Iasi, 700115 Iasi, Romania; diana-cristala.nitescu@umfiasi.ro (D.C.K.N.); liliana.pasarin@umfiasi.ro (L.P.); silvia.martu@umfiasi.ro (S.M.); alexandra-cornelia.c.oanta@umfiasi.ro (C.T.); bogdan.vasiliu@umfiasi.ro (B.V.); sorina.solomon@umfiasi.ro (S.M.S.)

[2] Dental Technology Department, Faculty of Dentistry, "Grigore T. Popa" University of Medicine and Pharmacy of Iasi, 700115 Iasi, Romania; ioana.martu@umfiasi.ro

\* Correspondence: ionut.luchian@umfiasi.ro; Tel.: +40-745-140-641

† Authors with equal contribution as the first author.

**Abstract:** Background: The aim of this study is to quantify chemotherapy agents in the saliva of oncology patients undergoing intravenous chemotherapy treatment, and evaluate the oral and periodontal clinical modifications and possible correlations between them. Materials and Methods: The study was conducted on 29 patients undergoing chemotherapy treatment with cisplatin, oxaliplatin or gemcitabine. Three total saliva samples were gathered from each patient in three key points regarding chemotherapy. The samples were then analyzed through methods of mass spectrometry and liquid chromatography. Results: Cisplatin and gemcitabine were only found in saliva at 30 min and 2 h after chemotherapy administration, however oxaliplatin was determined in all three samples. Clinically, the most accentuated clinical attachment loss and CPITN scores were observed on mandibular teeth, whereas the highest values for the Sillness and Loe gingival index and gingival bleeding index were in the lateral maxillary areas. We found no statistically significant correlation between the periodontal parameters and chemotherapy concentration in saliva. Conclusion: A fraction of systemically administered chemotherapy can also be found in the saliva of oncology patients and have the potential to exacerbate oral infections and cause local and systemic complications throughout the oncology treatment. Further research is required in order to fully understand how chemotherapy can influence the development of periodontal disease.

**Keywords:** chemotherapy; saliva; periodontal disease; cisplatin; oxaliplatin; gemcitabine

## 1. Introduction

Chemotherapy is a widely used treatment for a variety of cancer forms [1]. Its therapeutic action consists of damaging cancer cells, but also healthy cells that have a high turn-over rate, thus leading to side effects [2]. The majority of patients undergoing chemotherapy, up to 86%, have reported side effects during treatment and have shown cumulative toxic effects [3]. The most cited prevalent adverse effects include a variety of immediate and late signs of toxicity, such as fatigue, nausea, alopecia, immunosuppression, insomnia, gastric discomfort [4,5], drug resistance, and infertility [6]. Oral cavity side effects include mucositis, which is among the most prevalent, bacterial, fungal and viral infections, neurological alterations, dysgeusia, dental alterations, hyposialia and xerostomia, tendency for hemorrhage and osteonecrosis [7].

Platinum-based drugs, such as cisplatin and oxaliplatin, are often used in the treatment of malignancies due to their effectiveness, although they frequently induce severe, dose-limiting side effects, such as nephrotoxicity and neurotoxicity [8]. On the other hand,

gemcitabine, a potent and specific deoxycytidine analog, is relatively well tolerated and in fewer cases does it induce side effects, such as anemia, neutropenia, or neutropenic fever [9]. In plasma, cisplatin can be found in two forms: protein bound and free circulating, which represents the active form of the drug. Up to 90% of the administered cisplatin dose becomes bound and inactivated by plasmatic proteins [10]. Similar to cisplatin, oxaliplatin undergoes a series of biological transformations once it reaches the bloodstream and also splits into three fractions: total platinum, erythrocyte platinum, which is the protein bound form, and free oxaliplatin [11]. Gemcitabine is one of the most used drugs in oncology, ranking third worldwide. It represents the base treatment for pancreatic cancer [12] and a series of solid tumors, such as breast, ovary, pulmonary, and urinary bladder cancer [13–15].

Saliva is valuable fluid from a diagnostic standpoint, which is used in several circumstances due to its complex and varied composition, which is often tightly related to the general status. Is has been used several times previously as an alternative to blood testing, for DNA analysis [16] or quantitative and qualitative testing of various drugs. Due to these reasons and the non-invasive character of the determinations, especially for oncology patients, we have chosen to use salivary analysis in the present study.

Determining chemotherapy drug levels in saliva through spectrometry may offer useful information without additional risks [17]. Collecting saliva samples, however, can prove difficult due to hyposialia and xerostomia that may often occur in chemotherapy patients and it is a possible reason for the limited number of studies in the scientific literature on this subject.

The aim of this study was to evaluate the concentration of chemotherapy drugs administered systemically in saliva and analyze the oral and periodontal clinical modifications in correlation with the chemotherapy levels present in the salivary fluid.

## 2. Materials and Methods

The study was conducted on 29 patients admitted to the Oncology Clinic of Hospital Victoria in Iasi between October 2018–May 2019, aged from 43 to 80 years old.

Inclusion criteria for the present study: patients suffering of systemic cancer and currently undergoing chemotherapy. The exclusion criteria were: non-cancer patients, patients with infectious or inflammatory diseases affect the periodontal status (with the exception of systemic cancer), patients that have received periodontal treatment in the last 6 months or antibiotic/anti-inflammatory treatment in the past 3 months, pregnant patients or minors.

The total number of patients were split into three groups: patients receiving cisplatin (N = 5), oxaliplatin group (N = 18) and gemcitabine group (N = 6). All the collected data was compared between the three groups.

### 2.1. General and Periodontal Data Collection

The data collection procedure was performed by gathering the general information of patients, identification, data regarding the oncology diagnostic (localization, stage), data about the chemotherapy treatment (administered drug, reported side effects) and clinical periodontal data on Ramfjord teeth (16, 21, 24, 36, 41 and 44): Sillness and Loe gingival index (GI), which provides a periodontal overview of the clinical aspect and bleeding of gingival tissues, gingival bleeding index (GBI), which further quantifies the amount of bleeding in regards to the presence of inflammation, CPITN index which offers information of periodontal damage, as well as treatment needs, probing depth (PD) and clinical attachment loss (CAL), which offers an in-depth view of present periodontal damage.

The present study was approved by the Ethics Committee of 'Grigore T. Popa' University of Medicine and Pharmacy (Iasi, Romania). All patients were informed in regards to the procedures specific to the present study and signed written consent for participation.

*2.2. Saliva Sample Collection and Analysis through Mass Spectrometry and Fluid Chromatography*

The collection of saliva samples was performed on patients included in the study before the clinical examination and after a light rinse with water of the oral cavity. The saliva was collected in sterile recipients until at least 5 mL of total saliva was gathered. The collecting procedure was performed before the start of the current chemotherapy iv (T0), repeated at 30 min after chemotherapy administration was concluded (T1) and at 2 h (T2), obtaining a total of 3 samples per patient; each sample received was appropriately inscribed with the patient initials, time of collection and type of chemotherapy administered. It is to be noted that T0 indicates the end of the previous chemotherapy administration cycle. T1 was chosen as a collection time due to the fact that maximum concentrations of chemotherapy agents seem to be detected 30 min after administration according to the literature [18]. T2 reports the rate of decrease in the above-mentioned types of drug concentrations. All concentration values were comparatively evaluated in order to determine the pharmacodynamic curve of each chemotherapy agent included in this study.

The three chemotherapy agents used in this study were cisplatin (CIS), oxaliplatin (OXA) and gemcitabine (GEM) provided by the European Pharmacopoeia. In order to attain superior stability, the standard samples for calibration were kept at 4–8 °C for 5 days at most. Formic acid, metallic alcohol, and water originating from an Elga PureLab system, were used for the establishment of analytical conditions.

The determinations were performed on a Tripluquadrupol Access Max mass spectrometry system and chromatographic separation on Transcend TLX 1 trip system. The sample centrifugation was done on a Hattick 4 centrifuge. Chromatographic separation was performed on a Phenomenex Kinetex C18 chromatographic column of 50 mm in length, internal diameter of 4.6 mm and particle size of 5 μm. The development of the analytical method was done in two stages: establishing working conditions on the mass spectrometry system and chromatographic optimization.

*2.3. Statistical Analysis*

Data gathered were registered, stored and statistically analyzed. The statistic processing of collected data was done in SPSS 24.0 using the Kolmogorov-Smirnov test for samples, the Pearson correlation coefficient, the ANOVA test, Wilcoxon rank comparison test and linear regression. Results with a value of $p < 0.05$ were considered statistically significant.

**3. Results**

The highest frequency regarding the oncology diagnostic was colon cancer (34.2% of total cases) followed by pulmonary cancer (24.1%), pancreatic and rectal cancer (13.8%) and esophageal cancer and liposarcoma (6.9%). All patients were diagnosed with stage IV cancer which means advanced or metastatic cancer.

Oxaliplatin treated cases represented the highest frequency (62.1%), followed by gemcitabine (20.7%) and cisplatin (17.2%) as seen in Table 1.

A number of 22 out of 29 (75.9%) patients presented adverse effects after chemotherapy of which the most frequent were nausea (34.5%), loss of appetite, alopecia (27.6%), mucositis, xerostomia, vertigo, prickling sensations and fever (13.8%).

The highest values of GI score of 3 were present more frequently on tooth 16 (15.4%), followed by 21 (7.4%) and 41 (3.7%). Oppositely, the lowest values were registered on 36 where a score of 1 represented 20%, as observed in Figure 1.

**Table 1.** Lot distribution regarding chemotherapy treatment, dosage, frequency and cancer types.

| Chemotherapy Agent | Number of Cases (%) | Mean Dose (mg) | Frequency of Administrations (Days) | Cancer Types |
|---|---|---|---|---|
| Cisplatin | 5 (17.2) | 48 | 21 | Pulmonary |
| Oxaliplatin | 18 (62.1) | 181.44 | 14 (N = 5 cases) 21 (N = 8 cases) | Esophagus, Colon, Rectum, Pancreas |
| Gemcitabine | 6 (20.7) | 1699.33 | 7 | Pulmonary, Pancreas, Liposarcoma |

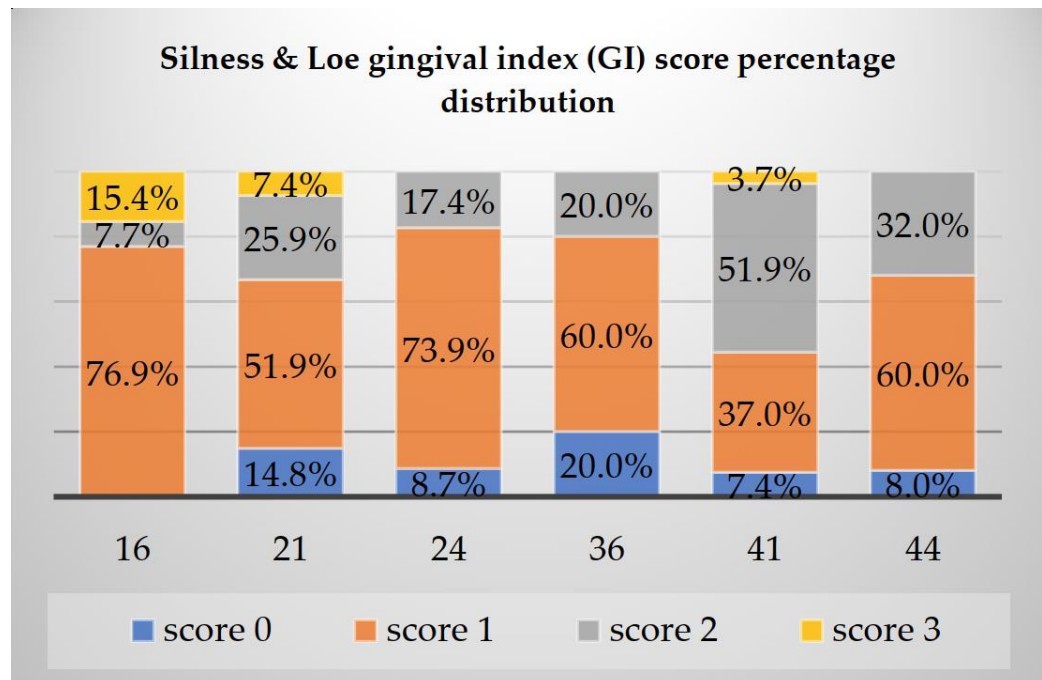

**Figure 1.** Sillness and Loe gingival index mean score percentages distributions on Ramfjord teeth (teeth numbers 16, 21, 24, 36, 41 and 44).

Similar to the GI results, the GBI showed the highest values at 16 (score 5–15.4%), followed by 21 (score 4–7.4%) and 41 (score 4–3.7%) and the lowest values were present at 36 with a score of 0—20%, as presented in Figure 2.

The CPITN index presented the highest values at 41 (score 3–14.8% and score 2–48.1%) and 44 (score 2–64%) with treatment needs of plaque and calculus removal, oral hygiene instructions and scaling and root planning.

The highest values of PD were found at 16 with a mean value of 1.611 mm and 36 with a mean value of 1.464 mm, thus the highest values were found in the lateral areas. Table 2 shows higher overall mean values in the maxilla rather than the mandible.

**Table 2.** PD values on Ramfjord teeth.

| Probing Depth (PD) on Tooth Number (Ramfjord) | N | Mean | Mean Standard Error | Standard Deviation | Minimum | Maximum |
|---|---|---|---|---|---|---|
| 16 | 13 | 1.6115 | 0.20165 | 0.72707 | 1 | 3.16 |
| 21 | 27 | 1.3844 | 0.11746 | 0.61035 | 0.58 | 2.91 |
| 24 | 23 | 1.3574 | 0.09628 | 0.46174 | 0.83 | 2.5 |
| 36 | 10 | 1.464 | 0.11756 | 0.37176 | 1.08 | 2 |
| 41 | 27 | 1.2863 | 0.14588 | 0.75802 | 0.58 | 2.83 |
| 44 | 25 | 1.2896 | 0.13483 | 0.67416 | 0.83 | 3.08 |

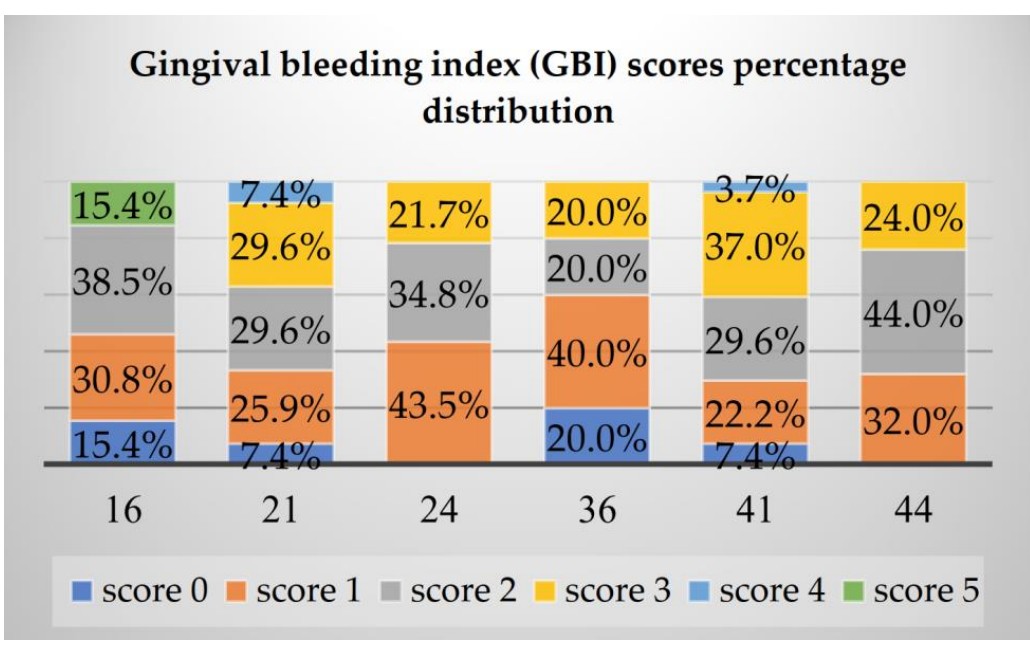

**Figure 2.** Gingival bleeding index mean score percentage distributions on Ramfjord teeth (teeth numbers 16, 21, 24, 36, 41 and 44).

CAL mean values, on the other hand, showed highest values on the mandible where 36, 41 and 44 presented mean CAL values of over 3 mm, as seen in Figure 3.

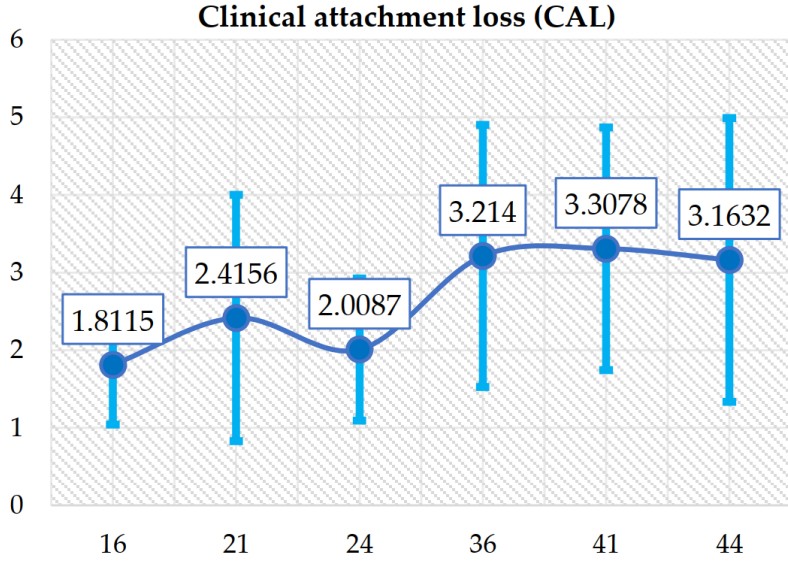

**Figure 3.** CAL mean value distribution on Ramfjord teeth.

Determining the concentration of chemotherapy in saliva showed a similar curve for all three agents included in this study, following the same pattern. Lowest concentrations were found in T0, cisplatin and gemcitabine being absent in the saliva samples (OXA T0 = 0.0272 µg/mL) and highest concentration values were found in T1 (CIS T1 = 0.0016 µg/mL, OXA T1 = 0.0507 µg/mL and GEM T1 = 0.0106 µg/mL). After two hours after administration, all three chemotherapy drugs showed a decrease in saliva concentration, as observed in Figure 4a–c.

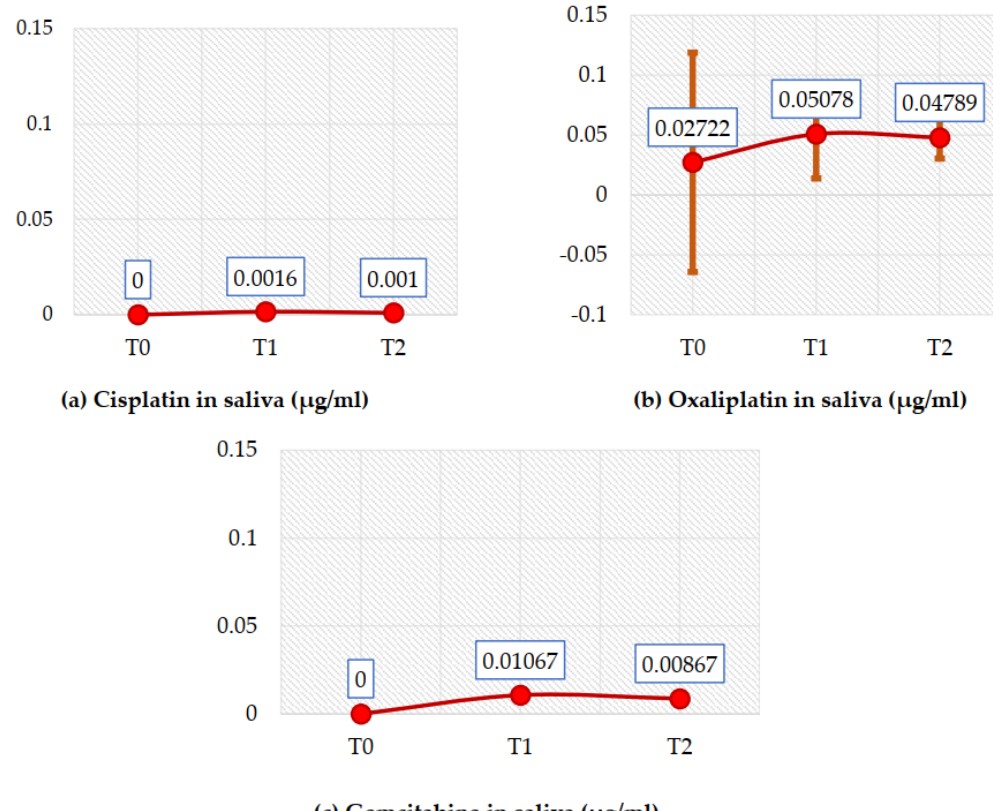

**Figure 4.** Mean values of chemotherapy drugs detected at T0, T1 and T2 in saliva. (**a**) Mean values of cisplatin concentration (μg/mL); (**b**) mean values of oxaliplatin concentration (μg/ml); (**c**) mean values of gemcitabine concentration (μg/mL).

Using the Wilcoxon test for ranked comparison we compared the overall chemotherapy concentrations in T0, T1, and T2 and obtained statistical significance, as shown in Table 3.

We then analyzed the correlation between the GI, GBI, CPITN, PD, and CAL and quantity of chemotherapy found in saliva in T0, T1 and T2 and found no statistical significance.

**Table 3.** Wilcoxon test for chemotherapy concentration comparison in T0, T1, and T2.

|  |  | N | Mean Rank | Rank Sum | Z Statistics | p Significance |
|---|---|---|---|---|---|---|
| Saliva chemotherapy in T1 compared with T0 | Negative ranks | 2 | 28.50 | 57.00 | −3.475 | **0.001, SS** |
|  | Positive ranks | 27 | 14.00 | 378.00 |  |  |
|  | Pairs | 0 |  |  |  |  |
|  | Total | 29 |  |  |  |  |
| Saliva chemotherapy in T2 compared with T0 | Negative ranks | 2 | 26.50 | 53.00 | −3.277 | **0.001, SS** |
|  | Positive ranks | 25 | 13.00 | 325.00 |  |  |
|  | Pairs | 2 |  |  |  |  |
|  | Total | 29 |  |  |  |  |
| Saliva chemotherapy in T2 compared with T1 | Negative ranks | 27 | 14.00 | 378.00 | −4.546 | **0.000, SS** |
|  | Positive ranks | 0 | 0.00 | 0.00 |  |  |
|  | Pairs | 2 |  |  |  |  |
|  | Total | 29 |  |  |  |  |

## 4. Discussion

Plasmatic determinations can be done for both cisplatin forms (protein bound or free circulating) [19], but also for the total plasmatic platinum quantity [20]. Plasma levels

of cisplatin become undetectable after 24–25 h post administration where the infusion duration was 0.5 h, as was the case in the present study, and highest concentrations in plasma were obtained in the first hour post administration the peak being at 30 min for both bound and free cisplatin [18].

The lowest concentration values determined in saliva in this present study were registered in T0, the highest were found to be in T1 right after administration at 30 min and then lowered again at 2 h (T2) after the administration was completed. These results confirm the presence of chemotherapy in the oral cavity and saliva in small quantities that follow a value curve similar to the plasmatic curve, and in the case of oxaliplatin it can still be detected in saliva even after a longer period of time, such as the end of the cycle between two administrations in our current study and coincides with our determinations in T0.

Clinical evaluation, which included the GI, GBI and CPITN indexes, alongside PD and CAL, partly reflected the level of hygiene and also offered a detailed view of the periodontal status in chemotherapy patients. Even though we did not achieve statistically significant results in correlating the presence of chemotherapy concentrations found in saliva with periodontal status, it has been stated in the literature that during chemotherapy there is a higher chance of infections occurring locally that can extend systemically [21,22]. Moreover, it has been proven that chemotherapy can influence the salivary immunoglobulins by decreasing IgG and IgA levels which in turn can partly explain the patient's susceptibility to oral infections [23]. The oral microbial community is influenced by chemotherapy as well as some saprophytic bacteria can become aggressive due to a decrease in granulocytes and increased fragility of the oral mucosa [24].

As necrotizing gingivitis is one of the more frequent forms of periodontal manifestations, it has been shown that certain bacteria, such as Prevotella, Fusobacterium, Actinobacillus, Actinomycetemcomitans and Actinomyces, are associated with infections occurring in patients receiving chemotherapy [25]. On the other hand, it was shown in the past that periodontal status can influence the microbial composition during chemotherapy [26].

The most important periodontal modification was observed on the mandible in our study, where the most extensive losses were registered at 36, 41, and 44 (mean CAL = 3.214, 3.308, and 3.163 mm, respectively). Radiotherapy has proven to have a similar effect, which registered 92% clinical attachment loss on the mandible; the loss being even greater in cases in which radiotherapy was aimed at this region [27]. Analyzing the CAL level offered additional data regarding the periodontal status of chemotherapy patients (overall mean value of CAL = 2.902 mm). From our observations and statistical analyses, we have determined that the progression pattern of periodontal disease in chemotherapy patients manifests more frequently with recessions rather than periodontal pockets.

The overall mean value of the Sillness and Loe GI was 1.265 in our study, which suggests the presence of minimal gingival modifications with the exception of a low number of patients, in whom a moderate or severe level of inflammation was observed. We also observed that the highest scores for the GI and GBI were found in the maxilla, especially in the lateral areas. During chemotherapy, the periodontal inflammation level is increased and, at the same time, so did the GI, which offers a general view on clinical periodontal status [28]. Under normal circumstances, the level of gingival affliction is dependent on bacterial plaque quantities and level of oral hygiene. It has been indicated in the literature that, during chemotherapy, periodontal inflammation can be exacerbated, even if the oral hygiene levels are good [29].

## 5. Conclusions

Our study demonstrates that a significant fraction of systemically administered chemotherapy can be found in the oral cavity and saliva of oncologic patients. Oxaliplatin is identified more easily in oral fluids compared to other studied chemotherapeutic



agents. The maximum concentrations for cisplatin, oxaliplatin, and gemcitabine were significantly quantified 30 min after the completion of administration.

Further research is required in order to determine the effect of chemotherapy on periodontal tissues and its impact on the prognosis of periodontal disease.

**Author Contributions:** D.C.K.N., I.M., S.M. and S.M.S. designed the study. C.T. and B.V. contributed substantially in drafting the work and revising it critically for important intellectual content. L.P. and I.L. contributed to the data analysis and data interpretation and edited the final form of the manuscript. All authors read and approved the final manuscript.

**Funding:** The study was partially funded by the 'Grigore T. Popa' University of Medicine and Pharmacy Iași-Romania, during the PhD studies of D.C.K.N.

**Institutional Review Board Statement:** The present study was approved by the Ethics Committee of 'Grigore T. Popa' University of Medicine and Pharmacy (Iasi, Romania). All protocols were in accordance with the provisions of the Declaration of Helsinki.

**Informed Consent Statement:** Informed consent was obtained from all subjects involved in the study.

**Data Availability Statement:** The data used and/or analyzed during the current study are available from the corresponding author on reasonable request.

**Acknowledgments:** I.M. contributed equally as the first author.

**Conflicts of Interest:** The authors declare no conflict of interest.

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
