# Peer review of "Determining Chemotherapy Agents in Saliva through Spectrometry and Chromatography Methods Correlated with Periodontal Status in Oncology Patients"

_applsci, doi:10.3390/app11135984_

Round 1

Reviewer 1 Report

The authors chose an interesting and important topic. However the manuscript requires some major corrections:

  1. The introduction is too general, with too less information about cancer and chemotherapy and its influence on the status of the oral cavity: dental manifestations of cancer in the oral cavity, a bit of background about pharmacodynamics of drugs used for this treatment (why these drugs were chosen for the research? Why there was only 2 hour time of evaluation, not longer?), finally short and long-term symptoms as well as complications of therapy in oncology patients shall be briefly described. Some parts of the text could be moved from discussion to the introduction.
  2. What was a control group? - healthy patients? Without chemotherapy (line 103 - what about other diseases?) Samples from how many patients?
  3. What was a distribution of drugs in the groups? Eg.: Group 1: Oxaliplatin - N=?  That shall be written in the material & methods chapter. 
  4. Rephrase inclusion and exclusion criteria: "Inclusion criteria:Patients with systemic cancer", Exclusion criteria: patients with the exception of systemic cancer" =means healthy? makes it confusing. 
  5. Why there was only 2 hour time of observation? What about long-term presence of drugs in saliva? How does periodontal situation changes after chemotherapy after eg. 21 days?  (line264). 21 days is the data from literature or authors result? - Lack of reference.
  6. The periodontal indexes were checked once at the begging before chemotherapy. Thus the periodontal oral status is a result of a disease/cancer itself not chemotherapy. They shall also be measured after and compared. It does not support the conclusion.
  7. What do numbers 1,2,3 mean on the FIg. 2 and 3? - shall be explained
  8. The results (no statistical significance) do not support the conclusion. 
  9. A statement that further investigation shall be provided! - in a conclusion 
  10. small corrections of language: separation (line 105), diagnostics (121), periodontal modifications (238), has (265)

Author Response

Thank you very much for your time and valuable remarks that helped us increase the quality of the manuscript. They were really useful and please kindly find below all the punctual modifications according to your recommendations:

  1. Two new paragraphs were added in the introduction that include information about cancer and chemotherapy as well as systemic and oral manifestations, short term and long term, with their respective references. A background of the chemotherapy agents used in this study was also provided as well as the motivation for their use (frequency of use and amount of side effects for all three).
  2. In the materials and methods section, the inclusion and exclusion criteria was reformulated in accordance with the given feedback.
  3. A distribution of the drugs in groups was added.
  4. Inclusion and exclusion criteria was rephrased
  5. A paragraph about the pharmacodynamics of the drugs was added in the introduction as well as a paragraph explaining the choice of saliva collection times. It was explained that T0 coincides with 21 days after the previous chemotherapy administration and also was mentioned in the materials and methods section that only patients with systemic cancer and undergoing chemotherapy was included in the study.
  6. Thank you for the observation. Indeed, the periodontal indexes were checked once before the collection of saliva, however I have made the corrections to make it clearer that the patients were not chemotherapy free, but inbetween chemotherapy administrations.
  7. Figure 2 and 3 have been explained.
  8. The conclusions have been modified to fit the results.
  9. A further investigation statement has been provided in the conclusion section.
  10. The language corrections have been addressed.

Again, thank you for the careful consideration of this manuscript.

Reviewer 2 Report

In the study by Niţescu et al., the authors quantified the levels of three different chemotherapeutic drugs in the saliva and evaluated the correlation between those levels and oral and periodontal parameters.

  1. Please, give a brief explanation about each of the clinical periodontal data on Ramfjord teeth and they measure.
  2. Please include a table with patients’ characteristics, such as type of cancer and treatment to substitute figure 1. It would be more interesting to see the data instead of a pie chart with the percentages of the chemotherapeutics.
  3. I did not understand figures 2 and 3 and what those results mean. Please be more specific. What does 16, 21, 24, etc mean?
  4. Sometimes the authors refer to chemotherapy quantity in the saliva. It should be concentration or levels and not quantity. Please correct that throughout the manuscript.
  5. The first line of the discussion the authors wrote “Chemotherapy presents multiple secondary effects and for this reason determining plasma levels have the potential to put the oncology patient at risk.” Why?
  6. These results confirm the presence of chemotherapy in the oral cavity and saliva in small quantities that follow a value curve similar to the plasmatic curve and in the case of oxaliplatin it can still be detected in saliva even after a longer period of time, like 21 days which represents the time between two administration in our current study.

Do not use “like” because it is too informal. Please re-write to “such as”

There is no reference for the statement about oxaliplatin being found in saliva after 21 years of chemotherapy administration. Is this from another study or is it a result from this one? Because the only time points presented here were 30min and 2h after chemotherapy administration.

  1. Line 238 “The most important periodontal modifications periodontal modification were…” there is a repetition of “periodontal modifications”.
  2. In the conclusion “correlating the concentration level in total 266 saliva of the chemotherapy drugs included in this study before, at 30 minutes and 2 267 hours were statistically significant.” Please be clearer. Does this mean that the difference in concentrations between time points were statistically significant?

Author Response

Thank you very much for your time and valuable remarks that helped us increase the quality of the manuscript. They were really useful and please kindly find below all the punctual modifications according to your recommendations:

  1. The meaning and uses of all indexes and Ramfjord teeth has been added.
  2. A table containing information regarding chemotherapy, dosage, frequency and cancer types has been added instead of the pie chart
  3. More information has been added in order to better explain the figures that are now numbered 1 and 2 instead of 2 and 3.
  4. All mentions of chemotherapy quantity have been corrected to chemotherapy concentrations.
  5. Introduction and discussions sections have been modified in order to better expose the subject matter.
  6. “21 days” has been corrected to “end of previous chemotherapy cycle” and an explanaition has been added in materials and methods that T0 is both the moment before the current chemotherapy administration and end of the previous chemotherapy cycle. Also, all “like” uses were corrected to “such as”.
  7. The repetition has been corrected in the text.
  8. Part of the conclusion “correlating the concentration level in total 266 saliva of the chemotherapy drugs included in this study before, at 30 minutes and 2 267 hours were statistically significant.” has been rephrased to be clearer.

Again, thank you for the careful consideration of this manuscript.

Round 2

Reviewer 1 Report

Thanks to the authors for the corrections. 

1. Write the proper name for the clinic with the capital letters (line 89)

Author Response

We have made the following correction:

  1. Line 89: "oncology clinic" has been modified to "Oncology Clinic"

Thank you kindly for your time.